# A Delay Performance Analysis and Wireless Resource Allocation Scheme Based on Martingale Theory

**DOI:** 10.3390/s25041164

**Published:** 2025-02-14

**Authors:** Baozhu Yu, Ziyang Jiao, Shuheng Xu

**Affiliations:** 1School of Information Science and Engineering, Shenyang Ligong University, Shenyang 110158, China; yubz19@mails.jlu.edu.cn; 2Beijing DualPi Intelligent Security Technology Co., Ltd., Beijing 100088, China; xushuheng@dualpi.com

**Keywords:** delay, QoS, martingale processes, aggregated traffic, wireless resource allocation

## Abstract

Statistical delay quality of service (QoS) provisioning is crucial for ultra-reliable low-latency communications (URLLCs). In this paper, a precise delay performance analysis framework is proposed based on martingale theory and a wireless resource configuration scheme is designed. A tight upper bound of delay violation probability is conducted for the aggregated traffic composed of bursty flows and independent identically distributed (i.i.d.) flows. A martingale of aggregated arrival processes is constructed. Based on the definition of martingale processes, the martingale parameters are determined by considering the statistical features of flows, which expose the impacts of heterogeneous flows entangled with each other on system delay. A stopping time event related to delay is defined. Leveraging the stopping time theory of martingale processes, the complementary cumulative distribution function of delay is captured, which reveals the implicit relationship among aggregated traffic, service schemes, and delay performance. Guided by the theoretical upper bound of delay violation probability, a bandwidth estimation algorithm is proposed, which facilitates the decoupling of the statistical delay QoS requirements as the bandwidth demands. Subject to the bandwidth demands, a wireless resource allocation problem is formulated. Based on the Lagrangian convex optimization framework, the closed form of the transmission power is obtained. Simulations verify the effectiveness of the martingale-based delay performance analysis method and power allocation scheme.

## 1. Introduction

In fifth-generation (5G) scenarios, ultra-reliable low-latency communications (URLLCs) services must exhibit a reliability of 99.999% with a delay of milliseconds [1], such as in the typical paradigm, the Internet of Things (IoT). Under the strict statistical delay quality of service (QoS) constraints, adequate resource allocation is the inevitable cost to guarantee reliability. More importantly, the data traffic generated from IoT devices possesses bursty or sporadic characteristics, which triggers an unimaginably large bandwidth requirement for the scant spectrum resources. This challenge is more acute in the uplink direction, since most of the IoT applications are uplink-centric by nature. To achieve a win–win situation between the delay QoS guarantee and efficient resource utilization, a precise theoretical analysis framework of the delay QoS is the bottleneck to guide the network configuration, which should reveal the relationships among the complex traffic features, the bandwidth provisioning, and the system delay.

Queuing theory provides the basic method to analyze the system delay. For the queuing systems with simple arrival and service processes, queuing theory presents many theoretical results about the average delay combined with the Little Law. In [2], a risk-resistant scheme for resource allocation to IoT traffic was proposed. Based on the M/G/1 queuing model, the packet latency target is realized by minimizing the average packet delay in violation cases, where “violation” is judged by introducing the “conditional value at risk” as a risk metric. In [3], an average queuing delay analysis framework was proposed for the non-orthogonal multiple access downlink system based on the queuing theory, which was utilized to obtain the optimum power allocation strategy. In [4], the authors applied the M/G/1 queuing model to analyze the erroneous transmission recovery delay of URLLC multi-user services, and based on this model, the maximum retransmission parameter of the Hybrid Automatic Repeat Request mechanism was designed to optimize the bandwidth requirement. To address the spectrum shortage issue for users, the authors in [5] utilized queuing models to further improve the average waiting time for users under two priority schemes. However, for arrival flows with complex stochastic features, this theory method exposes mathematical barriers. The statistical characteristics of delay are unable to be obtained, especially in the queuing systems where the traffic from the URLLC user equipment (UE), such as IoT, could be composed of multiple flows with heterogeneity. Meanwhile, in the QoS framework, the probability distributions or the cumulative distribution functions of delay are focused on, rather than the average value.

Stochastic network calculus (SNC) gained momentum as a mathematical framework in terms of statistical delay performance analysis. The departure process and the delay can be defined by the (min, +) convolutions. For IoT devices, a digital twin (DT)-assisted network slicing resource allocation scheme was proposed in [6]. SNC theory was used to analyze the end-to-end delay violation probability and characterize the relationship between delay and service reliability under given traffic arrival distributions and delay constraints. The theoretical results guided the construction of a joint resource allocation problem to maximize the utility of the infrastructure provider while guaranteeing the deterministic delay. In [7], the authors used network calculus theory to derive the end-to-end delay upper bound for service chain routing, and the relationship between the upper bound of delay and the resource allocation of virtual network function nodes was analyzed. In [8], the end-to-end delay of the industrial IoT traffic was evaluated based on SNC with moment-generating functions (MGFs), where the industrial wireless link was challenged by the complex fading channel and the multinode wired network was controlled by common schedulers. A method based on Meijer G-functions was proposed to calculate the MGF for the service processes of various wireless fading channels. In industrial IoT scenarios supported by visible-light communication, the authors in [9] analyzed the delay violation probability of data traffic using SNC. Based on this, a low-complexity resource slicing scheme was proposed and the LED layout characteristics was exploited to minimize the superframe duration. In the SNC framework, union bound inequality is the enabler, which converts the (min, +) convolutions into univariate convolution in classical algebra. Unfortunately, union bound inequality can induce loose upper bounds of delay violation probability [10], which could cause the wasting of resources if these results are used to guide the bandwidth allocation.

The effective bandwidth (EB) and effective capacity (EC) theories are classical methods for analyzing delay violation probability, i.e., the complementary cumulative distribution function of delay. These theories originated from large deviation theory [11]. To address the resource allocation of service function chains (SFC) under QoS constraints, the authors in [12] proposed a multi-SFC resource sharing model coupled with bandwidth allocation based on EB and EC theories. In [13], for emerging industrial IoTs, the reliability–latency trade-off of short packet transmission was studied. The effective bandwidth approach was adopted to obtain the delay violation probability given a hard delay constraint. An approximate but closed-form QoS exponent that bridged the delay violation probability of short packets and the error probability of finite-blocklength coding was derived. In [14], the effective capacity theory was adopted to express the latency constraint by introducing latency exponents of URLLCs. Focusing on the tail behavior of random latency experienced by packets, the authors performed spectrum and power allocation based on large-scale channel information. The authors in [15] designed a multi-user system of URLLCs that adopted finite-block-length (FBL) channel codes under QoS constraints. The effective capacity framework was employed to characterize the throughput by considering the delay requirements. The relationship between throughput and bandwidth was investigated. The EB and EC theories construct a delay performance analysis framework and prove the relationship between the arrival process, service process, and QoS index [16], which contributes to mapping multiple QoS requirements as one parameter. However, for bursty traffic, the processing pattern of flow approximation is adopted. If the queuing length of data in the buffer is short, EB and EC could provide a loose upper bound of delay violation probability [17].

Martingale theory has been verified as having superiority in delay QoS evaluation. The obtained upper bound of the delay violation probability improves state-of-the-art bounds by several orders of magnitude [18], especially for bursty traffic. The arrival and service processes can obtain modularized modeling by means of the martingale processes, facilitating the analysis of the delay performance of bursty traffic and aggregated traffic [19]. In [20,21], for the scenario where IoT and MTC devices coexist, differentiated ALOHA random access algorithms were proposed under the martingale-based delay violation probability bound constraints. In [22], the authors introduced the new concepts of arrival and service martingale processes and proposed a Sparrow Search Algorithm based on martingale theory to obtain the optimal service rate. Based on the strict delay violation probability bounds of martingale theory, the authors in [23] proposed a maximum capacity and optimal task allocation scheme under a given delay QoS constraint. Martingale theory also contributes to achieving end-to-end delay evaluation and network optimization. For URLLC applications, in [24,25], the authors proposed an reliability analysis framework for end-to-end delay in multi-hop systems. Based on the access state and channel characteristics in the wireless access network, they introduced a service rate optimization scheme for dynamic transmission power allocation. In [26], to guarantee the end-to-end delay QoS of the IoT traffic, the authors analyzed the end-to-end delay performance in edge computing scenarios and investigated two power-level time-slot ALOHA non-orthogonal multiple access schemes, and based on martingale theory, the authors rationalized resource allocation. The advantages of martingale theory in aiding the precise delay analysis can be summarized as follows. (1) The heterogeneous arrivals can be constructed as modularized martingale processes, which contribute to the modeling of aggregated traffic. (2) The latency process in a queuing system, the difference between two or more complex random processes, can be described in the martingale domain. (3) The strict inequality conclusions of martingale processes support the probability distribution analysis of delay. In 5G and 6G networks, the traffic with more complex statistical features needs to be focused on, such as arrival processes composed of multiple heterogeneous flows. The existing martingale-based delay analysis methods should be extended. In Table 1, the comparison between different theories on delay performance analysis is concluded. For the simple arrival flows, such as the one following Poisson distributions, EB/EC, SNC, and martingale theories can all provide precise theoretical analysis of delay violation probability. For the bursty flows and the aggregated traffic, these methods have limitations in terms of arrival models. Martingale theory has shown advantages in statistical delay analysis of bursty and aggregated traffic. In this paper, we expect to construct a novel delay performance analysis framework for aggregated traffic based on martingale theory. Relying on the theoretical results, a reasonable wireless resource allocation scheme is studied.

In this paper, a precise martingale-based statistical delay QoS analysis framework is proposed and a resource allocation algorithm is designed. A martingale process related to aggregated traffic is constructed, which consists of heterogeneous data flows. Based on the stopping time theory of martingale processes, a tight upper bound of delay violation probability is obtained. Guided by the theoretical upper bound, the bandwidth demands of the aggregated traffic are estimated, and a transmission power allocation problem is formulated. The closed form of power consumption is derived. The contributions of this paper can be summarized as follows:A martingale of aggregated arrival processes is constructed, where the aggregated traffic consists of bursty flows and independent identically distributed (i.i.d.) flows. The martingale parameters are determined based on the definition of martingale processes, considering the statistical characteristics of flows fully. The martingale of aggregated arrival processes provides a modularized description paradigm, which facilitates revealing the impacts of heterogeneous stochastic arrival processes entangled with each other on the system latency.A precise statistical delay analysis framework is proposed. The delay of aggregated traffic can be modeled in the martingale domain, relying on the martingale constructions of aggregated arrival processes and service processes. We define a stopping time event about delay violation. Leveraging the stopping time theory of martingale processes, a precise upper bound of delay violation probability is derived, which exposes the relationship among complex arrivals, bandwidth provisioning, and delay performance.A bandwidth estimation and wireless resource allocation scheme is proposed, guided by the theoretical upper bound. The bandwidth estimation algorithm is performed, which decouples the statistical delay QoS requirements and the bandwidth demands under the delay QoS requirements constraints. We design a dichotomous search algorithm to capture the required service rates for aggregated traffic. A physical layer resource allocation optimization problem is formulated subject to bandwidth demands. Using the Lagrange multiplier method, the closed form of transmission power is yielded.

The following sections are arranged in the following manner: in Section 2, the communication network model and queuing system model are introduced. In Section 3, the martingale processes of aggregated arrivals and service are constructed. In Section 4, based on the stopping time theory, we derive the tight delay violation probability bound for the aggregated traffic. In Section 5, the bandwidth estimation and wireless resource allocation algorithms are designed. In Section 6, simulation results are obtained and the results are interpreted. Finally, conclusions are presented in Section 7.

## 2. Network Model and Queuing System

The network model diagram is shown in Figure 1a. We consider an uplink system for the URLLC data traffic based on the 5G software-defined radio access network, which is composed of a centralized baseband server (BBS) pool, remote radio heads (RRHs), and user equipment (UE). RRHs are controlled by the baseband units in BBS through fronthaul links, which are near the UE to provide high data rates and reliability provisioning. The uplink between the UE and RRHs employs wireless transmission. The RRHs and UE are all single-antenna, and the RRH-centric clustering is performed. Multiple pieces of UE are associated with one RRH. Each RRH occupies the independent frequency band to avoid the inter-cell interference. In an RRH, the set of sub-channels is M={1,…,M}. We study the QoS guarantee in some cells. The UE is assumed to carry the aggregated traffic with strict delay QoS requirements. The aggregated traffic consists of data flows with heterogeneous statistical features. In the cover area of the RRH, *K* pieces of UE are randomly distributed within this range. The set of UE is denoted as K={1,…,K}. In the target cell, when pieces of UE generate data packets, the slotted ALOHA-like random access schemes are adopted, i.e., the grant-free access paradigm. Each device persistently transmits the data from its buffer over the sub-channel. Relying on the sporadic or bursty data pattern at each piece of UE, prompt ALOHA-like data transmission is expected to help devices flush their buffer soon after data generation. In this access scheme, each piece of UE directly sends the data packets to its closest RRH without scheduling. To reduce mutual interference, each piece of UE randomly and independently selects a sub-channel among the *M* available sub-channels for each transmission. The ALOHA-like random access scheme can be described by the Bernoulli distribution [21]. Each piece of UE has a successful access probability. It has been proven that the ALOHA-like random access scheme can offer lower average queue size and delay as long as the number of pieces of UE is low (a high successful access probability) [27]. Based on the Shannon theory, the achievable transmission rate of the UE *k*, if the access is successful, is given by(1)$$\begin{array}{c} R_{k}=w\log_{2}\left(1+\frac{p_{k}\left(n\right)h_{k}^{2}}{v^{2}}\right), \end{array}$$
where $p_{k}\left(n\right)$ is the transmission power of UE *k* at time slot *n*. $h_{k}^{2}$ is the channel gain of UE *k*, and $v^{2}$ is the noise power. *w* is the bandwidth allocated to UE *k*. The channel gain model of a piece of UE includes path loss, shadow fading, and small-scale fading. Path loss is related to the distance Dr between the user and the RRH. Shadow fading is assumed to follow a normal distribution, and small-scale fading follows a complex Gaussian random distribution. The increase in path loss means that the signal strength rapidly weakens as the distance increases, resulting in a decline in the signal quality at the receiver. Shadowing introduces spatial non-uniformity in the signal distribution, leading to substantial fluctuations in the received signal strength at the same location. This form of fading not only diminishes the average signal power but also amplifies the variability of the received signal, thereby increasing the level of uncertainty within the communication system. Consequently, it adversely affects both the stability of the transmission rate and the overall delay performance. Small-scale fading has a direct impact on transmission rate and delay, as it induces rapid variations in the signal. This can also lead to an increase in the signal recovery time, thereby resulting in higher delay.

For each piece of UE, the data transmission process is modeled as a queuing system, as shown in Figure 1b. The aggregated traffic is modeled as the arrival process, and the access scheme offered by the RRH is described as the service process. We assume that the arrival process consists of two types of random processes, which are Markov-modulated and i.i.d., respectively. These flows are independent of each other, and the aggregated arrival process is also independent of the service process. The buffer is assumed to be infinite. The first-in first-out rule is adopted in the buffer. Due to the stochastic features of arrival and service processes, queuing behavior could occur.

In UE *k*, the arrival processes of flow 1 and flow 2 are denoted as $\left\{a_{k,1}\left(n\right),n\geq 0\right\}$ and $\left\{a_{k,2}\left(n\right),n\geq 0\right\}$, respectively, where $a_{k,j}\left(n\right),j\in \left\{1,2\right\}$ is the number of data packets arriving in time slot *n*. Let $A_{k}\left(0,n\right)$ denote the accumulated arrival packets from slot 0 to *n* for the aggregated arrival flows and $A_{k}\left(0,n\right)=A_{k,1}\left(0,n\right)+A_{k,2}\left(0,n\right)$, where $A_{k,j}\left(0,n\right),j\in \left\{1,2\right\}$ represents the accumulated arrivals of the flow 1 and flow 2, respectively. The service rate, $s_{k}\left(n\right)$, in the unit of packets/slot, is the transmission rate provided for the UE *k*. Similarly, the accumulated service rates can be represented by $S_{k}\left(0,n\right)$. The observation of arrival packets is at the beginning of each time slot. The observation of departure packets and buffer backlog is at the end of a slot. As an example, Figure 1c shows the number of arrival, service, departure, and backlog packets in a UE with time slot evolution. The number of arrival packets at each slot is the aggregated of flow 1 and flow 2. Due to the randomness of arrival and service processes, the departure process is also stochastic. Based on SNC theory, the accumulated departure process of the aggregated traffic can be represented as [18](2)$$\begin{array}{c} D_{k}\left(0,n\right)=A_{k}\left(0,n\right)*S_{k}\left(0,n\right)=\underset{m\geq 0}{\inf}\left\{A_{k}\left(0,m\right)+S_{k}\left(m,n\right)\right\}, \end{array}$$
where the processes of arrival, service, and departure are connected by a (min, +) convolution.

The delay can be modeled as(3)$$\begin{array}{c} W_{k}\left(n\right)=\min\left\{l\geq 0|A_{k}\left(0,n-l\right)\leq D_{k}\left(0,n\right)\right\}. \end{array}$$

$W_{k}\left(n\right)$ is the horizontal distance between the curves $A_{k}\left(0,n\right)$ and $D_{k}\left(0,n\right)$, as shown in Figure 1d. For the aggregated arrival process, its delay should meet the probability constraints, which can be given as(4)$$\begin{array}{c} P\left\{W_{k}\left(n\right)\geq W_{\max}\right\}\leq \varepsilon , \end{array}$$
where $W_{\max}$ represents the delay target. ε denotes the delay violation probability threshold.

## 3. Martingale Constructions for Arrivals and Service Processes

In this section, we model the aggregated traffic and the service process in the martingale domain. A martingale of aggregated arrival processes is constructed based on the basic concept of martingale processes, and a martingale of service processes is introduced. The martingale constructions of the aggregated arrival processes and service processes enable the analysis of delay violation probability.

Firstly, the definition of a martingale process is presented in Definition 1.

**Definition 1** (Martingale processes)**.**
*$\left(\Omega ,F_{\infty },P\right)$ is a probability space and ${\left\{F_{n}\right\}}_{n\geq 0}$ is a filtration. A random process {X(n),n≥0} is adapted to ${\left\{F_{n}\right\}}_{n\geq 0}$. If {X(n),n≥0} meets (Equation 5) and (Equation 6),*

(5)
$$\begin{array}{c} E[|X(n)|]<\infty ,\forall n\geq 0, \end{array}$$


(6)
$$\begin{array}{c} E[X\left(n+1\right)|F_{n}]=X\left(n\right),\forall n\geq 0, \end{array}$$

*then {X(n),n≥0} is a martingale process [25].*


Based on the definition of martingale processes, we construct a martingale of the aggregated traffic, which facilitates the model of complex arrival processes in the martingale domain, as shown in Definition 2. The aggregated traffic consisting of Markov-modulated arrival processes and i.i.d. arrival processes is considered.

**Definition 2** (The martingale of aggregated arrival processes)**.**
*For the aggregated traffic $A_{k}\left(0,n\right)$ composed of $\left\{a_{k,j}\left(n\right),n\geq 0\right\},j\in \left\{1,2\right\}$, there exists $Ka_{k,j}\left(\theta_{k}\right)\geq 0$, and the function $ha_{k,j}\left(a_{k,j}\left(n\right),\theta_{k}\right):rng\left(a\right)\to R^{+}$. The random process $M_{k}^{a}\left(n;ha_{k,j}\left(a_{k,j}\left(n\right),\theta_{k}\right),\theta_{k},Ka_{k,j}\left(\theta_{k}\right)\right)$ is constructed as*

(7)
$$\begin{array}{c} M_{k}^{a}\left(n;ha_{k,j}\left(a_{k,j}\left(n\right),\theta_{k}\right),\theta_{k},Ka_{k,j}\left(\theta_{k}\right)\right)=\prod_{j}ha_{k,j}\left(a_{k,j}\left(n\right),\theta_{k}\right)e^{\theta_{k}\left(A_{k}\left(0,n\right)-n\sum_{j}Ka_{k,j}\left(\theta_{k}\right)\right)}, \end{array}$$

*meeting the definition of martingale processes. Then, $M_{k}^{a}\left(n;ha_{k,j}\left(a_{k,j}\left(n\right),\theta_{k}\right),\theta_{k},Ka_{k,j}\left(\theta_{k}\right)\right)$ is admitted as a martingale of aggregated arrival processes.*


$ha_{k,j}\left(a_{k,j}\left(n\right),\theta_{k}\right)$ is a function of the statistical features of $\left\{a_{k,j}\left(n\right),n\geq 0\right\},j\in \left\{1,2\right\}$ and the parameter $\theta_{k}$. $rng\left(a\right)\;$ represents the operator. $Ka_{k,j}\left(\theta_{k}\right)$ is a function of $\theta_{k}$. $M_{k}^{a}\left(n;ha_{k,j}\left(a_{k,j}\left(n\right),\theta_{k}\right),\theta_{k},Ka_{k,j}\left(\theta_{k}\right)\right)$ is related to $ha_{k,j}\left(•\right)$, $Ka_{k,j}\left(•\right)$ and $\theta_{k}$. Why (Equation 7) is a martingale process depends on the selection of parameters $K_{a_{k,j}}\left(\theta_{k}\right)$ and $ha_{k,j}\left(a_{k,j}\left(n\right),\theta_{k}\right)$. In this paper, for the heterogeneous flows with burstiness and i.i.d. features, the different martingale parameters are determined, as shown in Lemma 1.

**Lemma** **1.**
*The martingale of aggregated arrival processes is constructed as in Definition 2. If $\left\{a_{k,j}\left(n\right),n\geq 0\right\}$ is a Markov-modulated process, which models the bursty flows, it can be denoted as $a_{k,j}\left(n\right)=f\left(y\left(n\right)\right)$. {y(n),n≥0} is a Markov chain with finite state space $S=\left\{0,1,\ldots .,N_{max}\right\}$ and $f:S\to R^{+}$ is a deterministic function. Then, the accumulated arrival is $A_{k,j}\left(0,n\right):=\sum_{l=0}^{n}f(y(l))$. The state transition matrix is denoted by $T={\left[T_{i,l}\right]}_{N_{\max}\times N_{\max}}$. The exponential column change matrix is defined as $T^{\theta }={\left[T_{i,l}^{\theta }\right]}_{N_{\max}\times N_{\max}}$, where $T_{i,l}^{\theta }=T_{i,l}e^{\theta_{k}f\left(y\left(n+1\right)\right)}$ and $T_{i,l}$ is the element of T. The spectral radius of $T^{\theta }$ is defined as $sp\left(T^{\theta }\right)$. Then, $ha_{k,j}$ is defined as the right eigenvector corresponding to $sp\left(T^{\theta }\right)$. $ha_{k,j}\left(a_{k,j}\left(n\right),\theta_{k}\right)$ is the ${\left(a_{k,j}\left(n\right)\right)}_{th}$ element of $ha_{k,j}$. $Ka_{k,j}\left(\theta_{k}\right)$ is set as $Ka_{k,j}\left(\theta_{k}\right)=\log sp\left(T^{\theta }\right)/\theta_{k}$. If $\left\{a_{k,j}\left(n\right),n\geq 0\right\}$ is an i.i.d. process, then the moment generation function is $E[e^{\theta_{k}a_{k,j}\left(n\right)}]$. Correspondingly, the parameter is set as $Ka_{k,j}\left(\theta_{k}\right)=\log E\left[e^{\theta_{k}a_{k,j}\left(n\right)}\right]/\theta_{k}$, and $ha_{k,j}\left(a_{k,j}\left(n\right),\theta_{k}\right)$ is a constant.*


Based on the parameters given in Lemma 1, we prove why Formula (Equation 7) is a martingale process next.

**Proof.** We take two flows as an example. Flow 1 is a Markov-modulated arrival process and Flow 2 is an i.i.d. arrival process. It is obvious that $ha_{k,j}\left(a_{k,j}\left(n\right),\theta_{k}\right)<\infty$ and $Ka_{k,j}\left(\theta_{k}\right)<\infty$. Thus, $M_{k}^{a}\left(n;ha_{k,j}\left(a_{k,j}\left(n\right),\theta_{k}\right),\theta_{k},Ka_{k,j}\left(\theta_{k}\right)\right)<\infty$ holds. Formula (Equation 5) in Definition 1 can be met. Relying on the independence between flow 1 and flow 2, we can write that(8)$$\begin{array}{c} \begin{array}{c} \begin{array}{c} E\left[M_{k}^{a}\left(n+1;ha_{k,j}\left(a_{k,j}\left(n+1\right),\theta_{k}\right),\theta_{k},Ka_{k,j}\left(\theta_{k}\right)\right)|F_{n}\right]\\ =E\left[\prod_{j}ha_{k,j}\left(a_{k,j}\left(n+1\right),\theta_{k}\right)e^{\theta_{k}\left(A_{k}\left(0,n+1\right)-\left(n+1\right)\sum_{j}Ka_{k,j}\left(\theta_{k}\right)\right)}|F_{n}\right]\\ =E\left[ha_{k,1}\left(a_{k,1}\left(n+1\right),\theta_{k}\right)e^{\theta_{k}\left(A_{k,1}\left(0,n+1\right)-\left(n+1\right)Ka_{k,1}\left(\theta_{k}\right)\right)}|F_{n}\right]\\ \times E\left[ha_{k,2}\left(a_{k,2}\left(n+1\right),\theta_{k}\right)e^{\theta_{k}\left(A_{k,2}\left(0,n+1\right)-\left(n+1\right)Ka_{k,2}\left(\theta_{k}\right)\right)}|F_{n}\right]. \end{array} \end{array} \end{array}$$For flow 1, it can be derived that(9)$$\begin{array}{c} \begin{array}{c} \begin{array}{c} E\left[ha_{k,1}\left(a_{k,1}\left(n+1\right),\theta_{k}\right)e^{\theta_{k}\left(A_{k,1}\left(0,n+1\right)-\left(n+1\right)Ka_{k,1}\left(\theta_{k}\right)\right)}|F_{n}\right]\\ =e^{\theta_{k}\left(A_{k,1}\left(0,n\right)-\left(n+1\right)Ka_{k,1}\left(\theta_{k}\right)\right)}E\left[ha_{k,1}\left(a_{k,1}\left(n+1\right),\theta_{k}\right)e^{\theta_{k}a_{k,1}\left(n+1\right)}|F_{n}\right]\\ \overset{\left(a\right)}{=}e^{\theta_{k}\left(A_{k,1}\left(0,n\right)-\left(n+1\right)Ka_{k,1}\left(\theta_{k}\right)\right)}\sum_{y(n+1)\in S}ha_{k,1}\left(a_{k,1}\left(n+1\right),\theta_{k}\right)e^{\theta_{k}f\left(y\left(n+1\right)\right)}P\left\{y\left(n+1\right)|y\left(n\right)\right\}\\ \overset{\left(b\right)}{=}e^{\theta_{k}\left(A_{k,1}\left(0,n\right)-\left(n+1\right)Ka_{k,1}\left(\theta_{k}\right)\right)}\sum_{l\in S}ha_{k,1}\left(a_{k,1}\left(n+1\right),\theta_{k}\right)T_{i,l}^{\theta }\\ \overset{\left(c\right)}{=}e^{\theta_{k}\left(A_{k,1}\left(0,n\right)-\left(n+1\right)Ka_{k,1}\left(\theta_{k}\right)\right)}\left(ha_{k,1}T^{\theta }\right)\left(a_{k,1}\left(n\right)\right)\\ \overset{\left(d\right)}{=}e^{\theta_{k}\left(A_{k,1}\left(0,n\right)-\left(n+1\right)Ka_{k,1}\left(\theta_{k}\right)\right)}sp\left(T^{\theta }\right)ha_{k,1}\left(a_{k,1}\left(n\right),\theta_{k}\right)\\ \overset{\left(e\right)}{=}e^{\theta_{k}\left(A_{k,1}\left(0,n\right)-nKa_{k,1}\left(\theta_{k}\right)\right)}ha_{k,1}\left(a_{k,1}\left(n\right),\theta_{k}\right). \end{array} \end{array} \end{array}$$In (Equation 9), formula (a) holds based on the definition of conditional expectation and $a_{k,j}\left(n\right)=f\left(y\left(n\right)\right)$. (b) relies on y(n)=i, y(n+1)=l, $T_{i,l}=P\left\{y\left(n+1\right)=l|y\left(n\right)=i\right\}$ and $T_{i,l}^{\theta }=T_{i,l}e^{\theta_{k}f\left(y\left(n+1\right)\right)}$. In (c), $\left(ha_{k,1}T^{\theta }\right)\left(a_{k,1}\left(n\right)\right)$ is the ${\left(a_{k,j}\left(n\right)\right)}_{th}$ element of vector $ha_{k,1}T^{\theta }$. Based on the relationship between the spectral radius and the right eigenvector, i.e., $ha_{k,1}T^{\theta }=sp\left(T^{\theta }\right)ha_{k,1}$, formula (d) can be obtained. The last step is completed according to $Ka_{k,1}\left(\theta_{k}\right)=\log sp\left(T^{\theta }\right)/\theta_{k}$.For flow 2 with i.i.d. features, it can be derived that(10)$$\begin{array}{c} \begin{array}{c} \begin{array}{c} E\left[ha_{k,2}\left(a_{k,2}\left(n+1\right),\theta_{k}\right)e^{\theta_{k}\left(A_{k,2}\left(0,n+1\right)-\left(n+1\right)Ka_{k,2}\left(\theta_{k}\right)\right)}|F_{n}\right]\\ \overset{\left(a\right)}{=}ha_{k,2}\left(a_{k,2}\left(n\right),\theta_{k}\right)e^{\theta_{k}\left(A_{k,2}\left(0,n\right)-\left(n+1\right)Ka_{k,2}\left(\theta_{k}\right)\right)}E\left[e^{\theta_{k}a_{k,2}\left(n+1\right)}|F_{n}\right]\\ \overset{\left(b\right)}{=}ha_{k,2}\left(a_{k,2}\left(n\right),\theta_{k}\right)e^{\theta_{k}\left(A_{k,2}\left(0,n\right)-\left(n+1\right)Ka_{k,2}\left(\theta_{k}\right)\right)}E\left[e^{\theta_{k}a_{k,2}\left(n\right)}\right]\\ =ha_{k,2}\left(a_{k,2}\left(n\right),\theta_{k}\right)e^{\theta_{k}\left(A_{k,2}\left(0,n\right)-nKa_{k,2}\left(\theta_{k}\right)\right)}. \end{array} \end{array} \end{array}$$In (Equation 10), (a) is based on the definition of conditional expectation. (b) holds because of the i.i.d. characteristics. The last step relies on the definition of $Ka_{k,2}\left(\theta_{k}\right)=\log E\left[e^{\theta_{k}a_{k,2}\left(n\right)}\right]/\theta_{k}$.Combined with (Equation 9) and (Equation 10) in (Equation 8), we can write that(11)$$\begin{array}{c} \begin{array}{c} \begin{array}{c} E\left[M_{k}^{a}\left(n+1;ha_{k,j}\left(a_{k,j}\left(n+1\right),\theta_{k}\right),\theta_{k},Ka_{k,j}\left(\theta_{k}\right)\right)|F_{n}\right]\\ =E\left[ha_{k,1}\left(a_{k,1}\left(n+1\right),\theta_{k}\right)e^{\theta_{k}\left(A_{k,1}\left(0,n+1\right)-\left(n+1\right)Ka_{k,1}\left(\theta_{k}\right)\right)}|F_{n}\right]\\ \times E\left[ha_{k,2}\left(a_{k,2}\left(n+1\right),\theta_{k}\right)e^{\theta_{k}\left(A_{k,2}\left(0,n+1\right)-\left(n+1\right)Ka_{k,2}\left(\theta_{k}\right)\right)}|F_{n}\right]\\ =e^{\theta_{k}\left(A_{k,1}\left(0,n\right)-nKa_{k,1}\left(\theta_{k}\right)\right)}ha_{k,1}\left(a_{k,1}\left(n\right),\theta_{k}\right)ha_{k,2}\left(a_{k,2}\left(n\right),\theta_{k}\right)e^{\theta_{k}\left(A_{k,2}\left(0,n\right)-nKa_{k,2}\left(\theta_{k}\right)\right)}\\ =M_{k}^{a}\left(n;ha_{k,j}\left(a_{k,j}\left(n\right),\theta_{k}\right),\theta_{k},Ka_{k,j}\left(\theta_{k}\right)\right). \end{array} \end{array} \end{array}$$Formula (Equation 6) in Definition 1 holds. The proof is completed.   □

For the service process with i.i.d. features, the martingale of service processes can be constructed following a similar line, as shown in Definition 3.

**Definition 3** (The martingale of service processes)**.**
*For the service process $\left\{s_{k}\left(n\right),n\geq 0\right\}$, there exists $Ks_{k}\left(\theta_{k}\right)\geq 0$, $\theta_{k}>0$, and the function $hs_{k}\left(s_{k}\left(n\right),\theta_{k}\right):rng\left(s\right)\to R^{+}$. The random process $M_{k}^{s}\left(n;hs_{k}\left(s_{k}\left(n\right),\theta_{k}\right),\theta_{k},Ks_{k}\left(\theta_{k}\right)\right)$ is constructed as*

(12)
$$M_{k}^{s}\left(n;hs_{k}\left(s_{k}\left(n\right),\theta_{k}\right),\theta_{k},Ks_{k}\left(\theta_{k}\right)\right)=hs_{k}\left(s_{k}\left(n\right),\theta_{k}\right)e^{\theta_{k}\left(nKs_{k}\left(\theta_{k}\right)-S_{k}\left(0,n\right)\right)},$$

*meeting the definition of martingale processes. Then, $M_{k}^{s}\left(n;hs_{k}\left(s_{k}\left(n\right),\theta_{k}\right),\theta_{k},Ks_{k}\left(\theta_{k}\right)\right)$ is admitted as a martingale of service processes.*


The parameters $Ks_{k}\left(\theta_{k}\right)$ and $hs_{k}\left(s_{k}\left(n\right),\theta_{k}\right)$ can be determined according to the conclusion in [18].

## 4. Delay Analysis Based on Martingale Theory

Based on Definitions 2 and 3 proposed in Section 3, we construct an analysis framework of delay violation probability. A stopping time event about delay is defined, and a tight upper bound of delay violation probability is derived in this section.

**Theorem** **1.**
*The martingales of aggregated arrival processes and service processes are presented in Definitions 2 and 3, respectively. For any $W_{\max}$, the delay violation probability inequality holds:*

(13)
$$\begin{array}{c} P\left\{W_{k}\left(n\right)\geq W_{\max}\right\}\leq \frac{E\left[\prod_{j}ha_{k,j}\left(a_{k,j}\left(0\right),\theta_{k}^{*}\right)\right]E\left[hs_{k}\left(s_{k}\left(0\right),\theta_{k}^{*}\right)\right]}{\min\left\{\prod_{j}ha_{k,j}\left(a_{k,j}\left(n\right),\theta_{k}^{*}\right)hs_{k}\left(s_{k}\left(n\right),\theta_{k}^{*}\right)\right\}}e^{-\theta_{k}^{*}\sum_{j}Ka_{k,j}\left(\theta_{k}^{*}\right)W_{\max}}, \end{array}$$

*where $\theta_{k}^{*}=\sup\left\{\theta_{k}>0:\sum_{j}Ka_{k,j}\left(\theta_{k}\right)=Ks_{k}\left(\theta_{k}\right)\right\}.$*


**Proof.** Based on the definition of delay, we can obtain(14)$$\begin{array}{c} \begin{array}{c} P\{W_{k}\left(n\right)\geq W_{\max}\}\\ =P\{A_{k}\left(0,n-W_{\max}\right)\geq D_{k}\left(0,n\right)\}\\ \leq P\{A_{k}\left(0,n-W_{\max}\right)\geq \underset{0\leq m\leq n}{\inf}\left\{A_{k}\left(0,m\right)+S_{k}\left(m,n\right)\right\}\}\\ \leq P\left\{\underset{0\leq m\leq n-W_{\max}\leq n}{\sup}\left\{A_{k}\left(m,n-W_{\max}\right)-S_{k}\left(m,n\right)\right\}\geq 0\right\}\\ =P\left\{\begin{array}{c} \underset{0\leq m\leq n-W_{\max}\leq n}{\sup}\{A_{k}\left(m,n-W_{\max}\right)-S_{k}\left(m,n\right)+\left(n-m\right)Ks_{k}\left(\theta_{k}^{*}\right)\\ -\left(n-m-W_{\max}\right)\sum_{j}Ka_{k,j}\left(\theta_{k}^{*}\right)\}\geq \sum_{j}Ka_{k,j}\left(\theta_{k}^{*}\right)W_{\max} \end{array}\right\}. \end{array} \end{array}$$The last equation holds based on the definition of $\theta_{k}^{*}$. We introduce a stopping time event regarding delay, which is denoted as ST. The moment when ST occurs for the first time is defined as the stopping time *N*.(15)$$\begin{array}{c} ST=\{A_{k}\left(m,n-W_{\max}\right)-S_{k}\left(m,n\right)+\left(n-m\right)Ks_{k}\left(\theta_{k}^{*}\right)\\ -\left(n-m-W_{\max}\right)\sum_{j}Ka_{k,j}\left(\theta_{k}^{*}\right)\geq \sum_{j}Ka_{k,j}\left(\theta_{k}^{*}\right)W_{\max}\}. \end{array}$$(16)$$\begin{array}{c} N:=\min\left\{\begin{array}{c} n>0:A_{k}\left(m,n-W_{\max}\right)-S_{k}\left(m,n\right)+\left(n-m\right)Ks_{k}\left(\theta_{k}^{*}\right)\\ -\left(n-m-W_{\max}\right)\sum_{j}Ka_{k,j}\left(\theta_{k}^{*}\right)\geq \sum_{j}Ka_{k,j}\left(\theta_{k}^{*}\right)W_{\max} \end{array}\right\}. \end{array}$$Based on the time-shift features of martingale processes, we can obtain that(17)$$\begin{array}{c} \begin{array}{c} M_{k}^{a}\left(n-W_{\max};ha_{k,j}\left(a_{k,j}\left(n-W_{\max}\right),\theta_{k}\right),\theta_{k},Ka_{k,j}\left(\theta_{k}\right)\right)\\ =\prod_{j}ha_{k,j}\left(a_{k,j}\left(n-W_{\max}\right),\theta_{k}\right)e^{\theta_{k}\left(A_{k}\left(m,n-W_{\max}\right)-\left(n-W_{\max}-m\right)\sum_{j}Ka_{k,j}\left(\theta_{k}\right)\right)},\\ M_{k}^{s}\left(n;hs_{k}\left(s_{k}\left(n\right),\theta_{k}\right),\theta_{k},Ks_{k}\left(\theta_{k}\right)\right)=hs_{k}\left(s_{k}\left(n\right),\theta_{k}\right)e^{\theta_{k}\left(\left(n-m\right)Ks_{k}\left(\theta_{k}\right)-S_{k}\left(m,n\right)\right)}, \end{array} \end{array}$$
are also two martingale processes.Meanwhile, leveraging the fact that the product of martingale processes is a new product, it can be derived that(18)$$\begin{array}{c} \begin{array}{c} M^{W}\left(n\right)=\prod_{j}ha_{k,j}\left(a_{k,j}\left(n-W_{\max}\right),\theta_{k}^{*}\right)hs_{k}\left(s_{k}\left(n\right),\theta_{k}^{*}\right)\\ \times e^{\theta_{k}^{*}\left(A_{k}\left(m,n-W_{\max}\right)-\left(n-W_{\max}-m\right)\sum_{j}Ka_{k,j}\left(\theta_{k}^{*}\right)+\left(n-m\right)Ks_{k}\left(\theta_{k}^{*}\right)-S_{k}\left(m,n\right)\right)}, \end{array} \end{array}$$
is a martingale process, which is related to delay. Applying the stopping time theory to $M^{W}\left(n\right)$, we have $E\left[M^{W}\left(0\right)\right]=E\left[M^{W}\left(N\wedge n\right)\right]$, where n∧N=min{n,N}. Then,(19)$$\begin{array}{c} \begin{array}{c} E\left[\prod_{j}ha_{k,j}\left(a_{k,j}\left(0\right),\theta_{k}^{*}\right)\right]E\left[hs_{k}\left(s_{k}\left(0\right),\theta_{k}^{*}\right)\right]\\ \geq E\left[\begin{array}{c} \prod_{j}ha_{k,j}\left(a_{k,j}\left(N-W_{\max}\right),\theta_{k}^{*}\right)hs_{k}\left(s_{k}\left(n\right),\theta_{k}^{*}\right)\\ \times e^{\theta_{k}^{*}\left(A_{k}\left(m,N-W_{\max}\right)-\left(N-W_{\max}-m\right)\sum_{j}Ka_{k,j}\left(\theta_{k}^{*}\right)+\left(N-m\right)Ks_{k}\left(\theta_{k}^{*}\right)-S_{k}\left(m,N\right)\right)}1_{\left\{N\leq n\right\}} \end{array}\right]\\ \geq \min\left\{\prod_{j}ha_{k,j}\left(a_{k,j}\left(n\right),\theta_{k}^{*}\right)hs_{k}\left(s_{k}\left(n\right),\theta_{k}^{*}\right)\right\}e^{\theta_{k}^{*}\sum_{j}Ka_{k,j}\left(\theta_{k}^{*}\right)W_{\max}}P\left\{N\leq n\right\}. \end{array} \end{array}$$It is possible that $P\left\{N\leq \infty \right\}=P\{W_{k}\left(n\right)\geq W_{\max}\},n\to \infty$. Thus, the upper bound of delay violation probability can be obtained as in Theorem 1.   □

## 5. Bandwidth Estimation and Wireless Resource Allocation Algorithms

Based on the theoretical analysis result of delay violation probability, a bandwidth estimation algorithm is designed to measure the required service capacity under the QoS requirements constraints. Subject to the bandwidth demands, a wireless resource allocation problem is performed and the power consumption of each piece of UE can be obtained. The overall research framework of this paper is summarized in Figure 2.

Initially, a martingale of aggregated arrival processes is presented in Section 3 based on the communication network model and the queuing system. Subsequently, leveraging the stopping time property of martingale processes, a tight upper bound of delay violation probability for aggregated traffic is derived, which is shown in Section 4. Guided by this theoretical result, an optimization problem about bandwidth estimation is formulated. and the bandwidth demands of UEs are captured. Under the bandwidth demands constraints, a wireless resource allocation algorithm is designed. The closed form of transmission power is obtained. The bandwidth estimation and wireless resource allocation are introduced as follows.

### 5.1. The Bandwidth Estimation Algorithm

In this section, a bandwidth estimation algorithm is designed, which can decouple the statistical reliability requirements as the bandwidth demands. The bandwidth demands are the minimum service rate to meet the delay QoS requirements, considering the complex arrival characteristics of aggregated traffic. In the queuing system, we define bandwidth demands as *B*. *B* should provide a reliability guarantee. Thus, we can construct Problem (Equation 20).(20)$$\begin{array}{c} \begin{array}{c} \min_{B}\left|\varepsilon -\frac{E\left[\prod_{j}ha_{k,j}\left(a_{k,j}\left(0\right),\theta_{k}^{*}\right)\right]E\left[hs_{k}\left(s_{k}\left(0\right),\theta_{k}^{*}\right)\right]}{\min\left\{\prod_{j}ha_{k,j}\left(a_{k,j}\left(n\right),\theta_{k}^{*}\right)hs_{k}\left(s_{k}\left(n\right),\theta_{k}^{*}\right)\right\}}e^{-\theta_{k}^{*}\sum_{j}Ka_{k,j}\left(\theta_{k}^{*}\right)W_{\max}}\right|\\ s.t.B\geq 0 \end{array} \end{array}$$

The objective of (Equation 20) is to make the theoretical upper bound approximate to the delay violation probability threshold. Unfortunately, there is no obvious closed-form expression for *B* in the martingale domain. It influences the system delay performance via the martingale parameters $\theta_{k}^{*}$ indirectly. There are many resource management algorithms to guarantee delay QoS, such as the Maximum-Largest Weighted Delay First, the dynamic weighted proportional fair scheduling algorithm, the dichotomous search, and so on [22,28]. Considering the computational complexity, we employ the dichotomous search procedure to capture the bandwidth demands. The dichotomous search algorithm needs an upper bound and a lower bound of the search variable. The EB/EC theories yield a relatively loose upper bound of the delay violation probability. If this theoretical result is used to guide the bandwidth estimation, a relative large bandwidth could be considered. To narrow the search range, the bandwidth derived from the effective bandwidth theory can be utilized as a bandwidth upper limit. Given that the bandwidth demands must exceed the average arrival rate to ensure system stability, the average arrival rate can be selected as the lower limit.

The bandwidth estimation algorithm is shown in Algorithm 1.
**Algorithm 1** Bandwidth Estimation**Input:** the arrival parameters of ak,jn,n≥0,j∈{1,2}, delay QoS requirements Wmax and ε. The maximum tolerance δ1 and δ2.**Output:** Bandwidth demands *B*
  1:The martingale parameters hak,jak,j(n),θk, Kak,j(θk) and hsksk(n),θk are derived according to the statistical distributions of arrival processes and the service process respectively. Based on the theoretical result of delay violation probability from EB and EC theories, the search upper bound High is captured. The average arrival rates of the aggregated traffic is set as the search lower bound Low. l=1. Mid(1)=Low.  2:**while **Low<High** do**  3:     l=l+1  4:     Mid(l)=High+Low2  5:     Mid(l) is regarded as the service rate to calculate θk*.  6:      According to θk*, the martingale parameters hak,jak,j(n),θk* and hsksk(n),θk* are determined. And the martingale-based upper bound in Theorem 1, which is denoted as delay provisionally, is calculated.  7:     **if** absdelay−ε≤δ1&&absMid(l)−Mid(l−1)≤δ2 **then**  8:          B=Mid(l)  9:          break10:      **else**11:           **if** absdelay>ε **then**12:                Low=Mid(l)13:           **else**14:                High=Mid(l)15:           **end if**16:      **end if**17:  **end while**


First, input the relevant arrival and service parameters. Then, in the initialization phase, the EB/EC theory is used to calculate the effective bandwidth value, which is determined as the upper limit of the bandwidth demands, High, as shown in the first line. The average arrival is set as the lower limit, which is Low.

In the *l*th iteration, let Mid(l)=High+Low/2. Calculate the upper bound of delay violation probability based on the value of Mid(l) (as shown in line 6), which is denoted as delay provisionally. If delay is close to the delay violation probability threshold and the difference value of Mid(l) between two consecutive iterations is less than $\delta_{2}$, Mid(l) is selected as the bandwidth demand. Otherwise, High or Low is covered. The binary search algorithm is executed continuously until the bandwidth demand *B* is obtained (as shown in lines 2–17).

The computational complexity of the binary search algorithm is O(logΓ), where Γ is the length of the search region. Therefore, the complexity of the bandwidth estimation algorithm shown in Algorithm 1 is O(log(High−Low)), where High and Low represent the initial ceiling and floor of the search region, respectively.

### 5.2. The Wireless Resource Allocation Algorithm

Based on Algorithm 1, we convert statistical delay QoS requirements into the system bandwidth demands *B*. This bandwidth can then be treated as the required transmission rate, which guides the allocation of wireless resources. For *K* pieces of UE associated with an RRH, the optimization problem for transmission power allocation can be formulated as follows.(21)$$\begin{array}{c} \begin{array}{c} \min_{p(n)}\sum_{k\in K}p_{k}\left(n\right)\\ s.t.w\log_{2}\left(1+\frac{p_{k}\left(n\right)h_{k}^{2}}{v^{2}}\right)\geq \frac{\left\lceil B\right\rceil H}{T},\forall k\in K,\left(1\right)\\ \sum_{k\in K}p_{k}\left(n\right)\leq p_{\max},\left(2\right) \end{array} \end{array}$$
where $p\left(n\right)=[p_{1}\left(n\right),p_{2}\left(n\right),p_{3}\left(n\right)....p_{K}\left(n\right)]$ represents the transmission power set. $p_{\max}$ represents the maximum power of the RRH. *H* is the length of a packet and *T* is the duration of a slot. In the power optimization problem, the constraint (1) ensures that the transmission rate of each piece of UE is greater than or equal to the minimum bandwidth demands. The constraint (2) ensures that the sum of the transmission powers of all pieces of UE must be less than or equal to the total transmission power of the RRH. Problem (Equation 21) is a convex optimization problem which can be tackled by the Lagrangian Gradient Algorithm. The Lagrangian duality function can be formulated.(22)$$\begin{array}{c} L\left(p\left(n\right),\alpha ,\beta \right)=\sum_{k=1}^{K}p_{k}\left(n\right)+\sum_{k=1}^{K}\alpha_{k}\left(\frac{\left\lceil B\right\rceil H}{T}-w\log_{2}\left(1+\frac{p_{k}h_{k}^{2}\left(n\right)}{v^{2}}\right)\right)+\beta \left(\sum_{k=1}^{K}p_{k}-p_{\max}\right), \end{array}$$
where $\alpha =[\alpha_{k}]$ and β are the Lagrange multipliers. Then, the dual problem can be constructed as follows:(23)$$\begin{array}{c} \begin{array}{c} \max_{\alpha ,\beta }\min_{p(n)}L\left(p\left(n\right),\alpha ,\beta \right)\\ s.t.\alpha_{k}\geq 0,\forall k\in K,\\ \beta \geq 0. \end{array} \end{array}$$

We use the Lagrangian Gradient Algorithm to solve Problem (Equation 23). Therefore, in the *l*th iteration, the Lagrange multipliers are ${\alpha_{k}}^{new}\left(l\right)$ and $\beta^{new}\left(l\right)$, respectively. The transmission power in the *l*th iteration can be expressed as(24)$$\begin{array}{c} p_{k}^{new}\left(l\right)=\max\left(\frac{\alpha_{k}^{new}\left(l\right)w}{\ln2(1+\beta^{new}\left(l\right))}-\frac{v^{2}}{h_{k}},0\right). \end{array}$$

For ${\alpha_{k}}^{new}\left(l\right)$ and $\beta^{new}\left(l\right)$, they can be further formulated as follows:(25)$$\begin{array}{c} \alpha_{k}^{new}\left(l+1\right)=\max\left(0,\alpha_{k}^{new}\left(l\right)+\eta \left[\frac{\left\lceil B\right\rceil H}{T}-w\log_{2}\left(1+\frac{p_{k}^{new}\left(l\right)h_{k}}{v^{2}}\right)\right]\right)\\ \beta_{k}^{new}\left(l+1\right)=\max\left(0,\beta_{k}^{new}\left(l\right)+\eta \left(\sum_{k=1}^{K}p_{k}\left(n\right)-p_{\max}\right)\right), \end{array}$$
where η is the learning rate.

The wireless resource allocation algorithm is shown in Algorithm 2.
**Algorithm 2** Wireless Resource Allocation**Input:** bandwidth *B*, hk, v2, *w*, *T*, *g*, pmax, the maximum tolerance tol.**Output:** The transmission power set.
  1:Based on the models of Small-scale fading, shadow fading, and user distance, the channel gains for UEs are determined.  2:**while **l>0** do**  3:     Construct the lagrange function L(p(n),α,β).  4:     Calculate αknew(l),k∈K and β.  5:     Get the lagrange multipliers αknew(l+1)=αknew(l)+η[BHT−wlog2(1+pknew(l)hkv2)] and βknew(l+1)=βknew(l)+η(∑k=1Kpk(n)−pmax).  6:     Update the transmission power pknew(l+1)=max(αknew(l+1)wln2(1+βnew(l+1))−v2hk,0).  7:     **if** |pknew(l)−pknew(l−1)|<tol) **then**  8:          pk(n)=pknew(l)  9:          break10:    **else**11:         l=l+112:    **end if**13:**end while**


First, according to Algorithm 1, input the solved bandwidth demands and the parameters related to the channel model. Then, construct the related optimization problem using the Lagrangian Gradient Algorithm (as shown in line 1).

Then, in the *l*th iteration, update the Lagrange multipliers and the transmission power based on Formulas (Equation 24) and (Equation 25) (as shown in lines 5–6). Check if the solved power satisfies the iterative stop condition. If it does, output the power. If not, rerun the program to line 2.

By constructing the Lagrangian dual problem, the tolerance for determining whether the power allocation result has converged (i.e., iteration stops) is tol. The tolerance is used to assess whether the difference between the current power allocation result and the previous iteration result is sufficiently small. If the difference is smaller than the tolerance, the result is considered converged, and the iteration stops. Because *K* pieces of UE are associated with the RRH. The computational complexity of Algorithm 2 is $O(\frac{K}{tol^{2}})$ [29].

## 6. Simulations and Results Analysis

In the simulation, since MATLAB supports a wide range of application scenarios, including signal processing, image processing, and communication system simulation, it provides enhanced computational performance, particularly when handling large-scale data. Therefore, the MATLAB 2023a version is adopted for this study. The commonly used variables without special emphasis are summarized in Table 2.

The effectiveness of the proposed bandwidth estimation and wireless resource allocation scheme is verified. A queuing system is simulated, where the number of pieces of UE is K=3. Each piece of UE carries the aggregated traffic, which consists of a Markov-modulated ON OFF (MMOO) flow and a Poisson flow.

In UE *k*, arrival flow 1 is modeled as a Poisson process, which is denoted as $\left\{a_{k,1}\left(n\right),n\geq 0\right\}$, and $a_{k,1}\left(n\right)$ is the number of data packets arriving in time slot *n*, which follows(26)$$P\left(a_{k,1}\left(n\right)=r\right)=e^{-\lambda }\frac{\lambda^{r}}{r!},r=0,1,2,\cdots ,$$
where λ is the average arrival rate. The Poisson process possesses the i.i.d. features. Thus, based on Lemma 1, the martingale parameters can be derived as follows:(27)$$\begin{array}{c} Ka_{k,1}\left(\theta_{k}\right)=\frac{\log E[e^{\theta_{k}a_{k,1}\left(n\right)}]}{\theta_{k}}\\ =\frac{\log\left(\sum_{m}P\left(a_{k,1}\left(n\right)=m\right)e^{\theta_{k}m}\right)}{\theta_{k}}\\ =\frac{\log\left(e^{-\lambda }\sum_{m}\frac{\lambda^{m}}{m!}e^{\theta_{k}m}\right)}{\theta_{k}}\\ =\frac{\log\left(e^{-\lambda }\sum_{m}\frac{{\left(\lambda e^{\theta_{k}}\right)}^{m}}{m!}\right)}{\theta_{k}}\\ =\frac{\log\left(e^{-\lambda }e^{\lambda e^{\theta_{k}}}\right)}{\theta_{k}}\\ =\frac{\lambda (e^{\theta_{k}}-1)}{\theta_{k}}. \end{array}$$

The parameter $ha_{k,1}\left(a_{k,1}\left(n\right),\theta_{k}\right)$ is set as $ha_{k,1}\left(a_{k,1}\left(n\right),\theta_{k}\right)=1$.

Arrival flow 2 is modeled as a MMOO process, which is denoted as $\left\{a_{k,2}\left(n\right),n\geq 0\right\}$. The model is shown in Figure 3. The state space is (0, 1). The state transition probabilities are $p_{a}$ and $p_{b}$. $p_{a}$ represents the probability of transitioning from the 0 state to 1 state between two adjacent time slots, and $p_{b}$ is the opposite. *R* denotes the arrival rate of the packet when it is in the 1 state. The state transition matrix can be denoted as T and the exponential column change matrix is $T^{\theta }$,(28)$$T^{\theta }=\left[\begin{array}{cc} 1-pa & pa\times e^{\theta_{k}R}\\ pb & \left(1-pb\right)\times e^{\theta_{k}R} \end{array}\right].$$

Based on Lemma 1, the martingale parameter $Ka_{k,2}\left(\theta_{k}\right)=\log sp\left(T^{\theta }\right)/\theta_{k}$. $ha_{k,2}$ is the right eigenvector with two elements. It satisfies $ha_{k,2}T^{\theta }=sp\left(T^{\theta }\right)ha_{k,2}$.

In this paper, the i.i.d. service process is considered. If the service rate is a constant *C*, then the martingale parameters also follow Definition 3. That is, $Ks_{k}\left(\theta_{k}\right)=C$ and $hs_{k}\left(s_{k}\left(n\right),\theta_{k}\right)=1$.

In the communication network scenario, the pieces of UE carry the data flows from URLLC services, which require strict delay QoS provisioning $P\left\{W\geq 1ms\right\}\leq 10^{-5}$. Thus, we assume that each piece of UE occupies a sub-channel exclusively to guarantee the strict QoS requirements. In the channel, the path loss model is given as $127+30\log_{10}\left(Dr\right)$ dB, where Dr is the distance between the UE and the RRH. The shadow fading follows a standard normal distribution with a standard deviation of 8, and the small-scale fading follows a complex Gaussian distribution with a standard deviation of 2. The bandwidth demands of each piece of UE are obtained by Algorithm 1, which is the constraint of the power allocation in problem (Equation 21). The transmission power is yielded and the service rate of each piece of UE is determined. The latency of $10^{6}$ slots is recorded. The ratio of the number of slots in which delay exceeds the delay target is regarded as the delay violation probability. The simulations are run 100 times. The simulated results of delay violation probabilities of each piece of UE are shown in the form of a box plot in Figure 4.

The top and bottom of each box plot represent the maximum and minimum values of the delay violation probability, respectively. The middle line of each box plot represents the median of the simulation results. It can be concluded that, when the delay target is set to eight slots (corresponding to 1 ms), the delay violation probabilities for all three pieces of UE are consistently below the given threshold $\varepsilon =10^{-5}$. The resource allocation method proposed in this paper could achieve statistical delay QoS provisioning. In our scheme, the bandwidth demands of aggregated traffic can be captured precisely, so that the power consumption can be measured reasonably. It is also implied that the martingale-based upper bound of delay violation probability for aggregated traffic is tight.

To analyze the performance of the bandwidth estimation algorithm based on martingale theory, a queuing system is simulated. The arrival process is composed of a Poisson process and an MMOO process. Service rates of the queuing system are the bandwidth demands obtained by Algorithm 1. The simulated results of delay violation probabilities are shown in Figure 5. Under the constraint of the delay targets, the delay violation probability is consistently below the given delay violation probability threshold. This further proves that the bandwidth estimation scheme proposed in this paper can reasonably estimate the bandwidth demands of aggregated traffic. Based on the theoretical upper bound of delay violation probability, Algorithm 1 could map the statistical QoS requirements as the required service rates.

Based on the simulation in Figure 5, a comparative analysis between martingale theory and the existing EB/EC theory is conducted. The upper bound of delay violation probability based on EB/EC theory [11] is used to guide the bandwidth estimation and resource allocation, and the delay violation probabilities are simulated. The results are shown in Figure 6. From the simulated results, the delay violation probability values of these two cases are less than the required delay violation probability threshold ($10^{-3}$). The box-plot of EB/EC is situated below the martingale theory, which can suggest that the bandwidth demands evaluated by the EB/EC theory is over-saturated and a suboptimal use of resources. It is also implied that the upper bound based on martingale theory is tighter than the one based on EB/EC theory, although the gap between the actual delay violation probability and $10^{-3}$ is also obvious in the case of strict delay targets (one to four slots). The martingale-based analysis method provides a relatively loose upper bound when the delay targets are small. The differences between the average of simulated delay violation probabilities and $10^{-3}$ are shown in Table 3. It can be demonstrated visually that for larger delay targets, the martingale-based simulated results match the QoS requirements well. For URLLC applications, such as IoT data, the delay target is at the millisecond level. Benefiting from the shorter duration of a slot in 5G new radio (0.125 ms), the corresponding delay target is 8 slots. Therefore, from Table 3, we can conclude that martingale theory is a better choice to support bandwidth demand evaluation.

The impact of system load on bandwidth demands is analyzed. In this scenario, the MMOO arrival parameters are fixed, and the average arrival rates of the Poisson flow are set as λ=3, λ=5 and λ=7, respectively. The results are shown in Figure 7. For relatively larger arrival parameters, the bandwidth demands for the system will be increased, while they decrease as the delay targets rise. The bandwidth demands fluctuate obviously in the case of small delay targets, while they change more slowly with the delay target relaxing. For loose delay QoS requirements, the impacts of i.i.d. statistical features in the aggregated traffic on resource allocation are controllable.

As the types of traffic become more diversified, the impact of bursty traffic on the system bandwidth demands is analyzed. Taking the MMOO arrival process as an example, the burstiness is usually measured by the squared coefficient of variation ($c^{2}$) [22].(29)$$c^{2}=\frac{p_{b}\left(2-p_{a}-p_{b}\right)}{{\left(p_{a}+p_{b}\right)}^{2}}.$$

A larger $c^{2}$ means a more violent burstiness. The burstiness of the Markov process is controlled by setting $c^{2}=3$, $c^{2}=5$ and $c^{2}=7$. With different values of $c^{2}$, the bandwidth demands versus delay targets are shown in Figure 8.

When burstiness is fixed, the bandwidth demands decrease with the delay target. By varying the burstiness of the MMOO process, for traffic with sharper burstiness, the system bandwidth demands intensify. This is because the bursty MMOO flow could trigger more complex queuing behavior. The delay QoS can only be provisioned at the cost of larger bandwidth. Thus, during bandwidth estimation and resource allocation, the low-dimensional statistical features of arrival flows, such as peak rates and average rates, are just the basic considered parameters. The delay performance is sensitive to the burstiness, meaning that the bandwidth demands are impacted indirectly.

The delay QoS is characterized by a pair of parameters $\left(W_{max},\varepsilon \right)$. In the analysis of arrival parameters, the impact of the delay target $W_{max}$ on the system bandwidth demands is discussed. Next, the impact of the delay violation probability threshold ε is evaluated. Three different delay violation probability thresholds, $\varepsilon =10^{-3}$, $\varepsilon =10^{-4}$, and $\varepsilon =10^{-5}$, are set. The results of bandwidth demands with different values of ε are shown in Figure 9. Under the same delay target $W_{max}$ constraint, a smaller value of ε corresponds to larger bandwidth demands, which means stricter QoS requirements. The system has less tolerance for delay violations, and thus requires more bandwidth to meet the delay QoS. The average arrival load in this case is 3.83 packets/slot. The bandwidth demands are more than double the average arrival of the aggregated traffic, even if the delay target is relaxed to 10 slots and $\varepsilon =10^{-3}$. This means that for the traffic with complex characteristics, the average feature is not enough to guide the bandwidth estimation and resource allocation.

Under the bandwidth demands constraints, the transmission power is optimized. Algorithm 1 is performed to capture required service rates. The power allocation optimization problem is formulated and tackled by the Lagrangian Gradient Algorithm. The proposed Algorithm 2 is compared with the Differential Evolution Algorithm and the Genetic Algorithm in terms of obtained power consumption, throughput, and convergence speed. The optimized transmission powers based on these three algorithms are shown in Figure 10. The achievable transmission rates corresponding to different delay targets provided by the proposed algorithm and the comparison algorithms are presented in Figure 11. The convergence performance of these three algorithms is simulated and the results are shown in Figure 12. The distance to the RRH impacts the resource consumption clearly when the channel models are assumed to be same between pieces of UE, as seen in Figure 10 and Figure 12. To provision the identical delay QoS requirements, UE further from the RRH requires higher transmission power. The distance affects path loss. UE at a greater distance experiences higher signal power loss during transmission.

In Figure 10, the transmission power captured by the Genetic Algorithm is larger than that based on Algorithm 2. From the results in Figure 4, it is verified that the allocated power provided by Algorithm 2 can support the delay QoS guarantee of each piece of UE. Thus, the solution by the Genetic Algorithm, a heuristic search algorithm paradigm, is higher than the actual need. This comparison algorithm could only present a feasible solution of power consumption, rather than the optimal solution. This method tends to overuse power to guarantee the statistical reliability requirements, which will cause the resource wasting. The Differential Evolution Algorithm could capture the approximate results compared with Algorithm 2. However, the convergence speed is slower than the Lagrangian Gradient Algorithm, as shown in Figure 12. The computation complexity of the Differential Evolution Algorithm is relatively high. Based on these three optimization algorithms, the searched transmission powers can all converge to a steady state. Although the convergence speed of the Genetic Algorithm is impressive, the obtained results of this algorithm need to make the network pay greater resource costs.

The objective function of the optimization problem that we formulated aims to minimize the total power consumption of all pieces of UE. Correspondingly, the total throughput, which is measured by the sum of the achievable transmission rates in this paper is calculated and shown in Figure 11. With the delay targets relaxing, the resource consumption should present a downward trend. Because these two algorithms used for comparison can only provide feasible solutions of transmission power, the waste of resources may occur, especially in the Genetic Algorithm. For the current delay QoS requirements of the aggregated traffic, excessive throughput is not necessary. The proposed Algorithm 2 can reduce resource usage effectively while guaranteeing the delay QoS requirements. According to this, the Lagrangian convex optimization framework is adopted in this paper.

The bandwidth demands versus transmission power are analyzed and the results are shown in Figure 13. As bandwidth demands increase, UE requires higher transmission power to meet QoS requirements. At the same time, for the same bandwidth demands, UE located farther from the RRH also needs more transmission power. In our proposed scheme, the statistical delay QoS requirements are mapped as the required service rates. As long as the arrival model and the QoS requirements remain constant, the bandwidth demands could be fixed, which contributes to guiding the physical layer design, especially in the complex channel environment.

To verify the impact of randomness of the service process on delay performance, the service obeying a Bernoulli distribution is set as an example. The successful access probability of a piece of UE is ps and the service rate is *C*. Based on Definition 3, the martingale parameters can be set as $Ks_{k}\left(\theta_{k}\right)=-\log\left(1-ps+pse^{-\theta_{k}C}\right)/\theta_{k}$, and $hs_{k}\left(s_{k}\left(n\right),\theta_{k}\right)=1$. The arrival process consists of an MMOO and Poisson aggregated traffic. The impact of access probability on the delay violation probability is analyzed. The simulated results are shown in Figure 14. It can be seen that as the delay target increases, the probability of delay violation gradually decreases. For lower access probabilities, the probability of exceeding the delay target is relatively larger. The delay performance is sensitive to the successful probability. When ps=0.5, the delay violation probability is in the order of $10^{-1}$. The system is unable to guarantee any delay QoS requirements. In this case, the UE cannot access the network frequently enough to clear the buffer, meaning that large transmission power allocation is not achieved. Thus, for URLLC devices, such as in the IoT, higher successful access probability should be provisioned. Only in this way can the ALOHA-like random access scheme provide the greatest advantages in terms of delay reduction.

## 7. Conclusions

In this paper, we propose a bandwidth estimation algorithm and a physical layer resource allocation scheme for URLLCs. Based on martingale theory, a precise statistical delay QoS analysis method is proposed. A martingale of aggregated traffic is constructed, which consists of Markov-modulated flows and i.i.d. flows. Based on stopping time theory of martingale processes, a tight upper bound of delay violation probability is obtained. Leveraging the theoretical result, the bandwidth estimation algorithm is performed, which maps the statistical delay QoS requirements as the bandwidth demands of the aggregated traffic. Finally, we formulate an optimization problem with the goal of minimizing the sum of UE powers. The optimization is subject to bandwidth demands and the total system transmission power. Utilizing the convex optimization theory, the required transmission power of UE is captured, thereby allocating resources efficiently and improving resource utilization.

The proposed theoretical analysis method is applicable to the aggregated traffic with Markov-modulated flows and i.i.d. flows. The state transition matrix and the moment generation function of flows are necessary for the framework. Thus, we need data modeling and parameter fitting technologies to support this process. In the future, for more complex traffic without explicit models, we should study how to analyze the delay performance based on martingale theory according to the limited sample data.

## Figures and Tables

**Figure 1 sensors-25-01164-f001:**
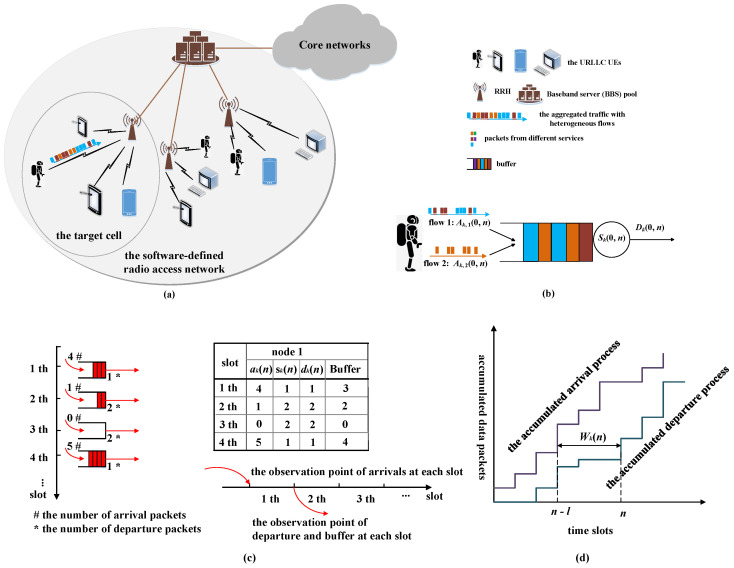
Scenario diagram. (**a**) is the network scenario. (**b**) is the corresponding queuing model. (**c**) is a diagram of data packets transmission. (**d**) is an illustration of the delay.

**Figure 2 sensors-25-01164-f002:**
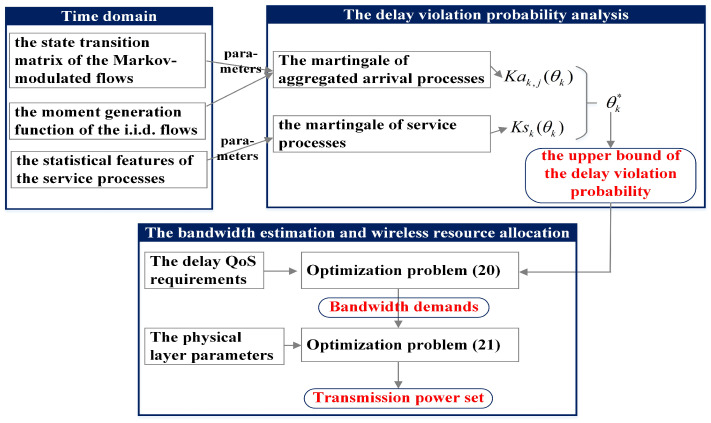
The overall framework.

**Figure 3 sensors-25-01164-f003:**
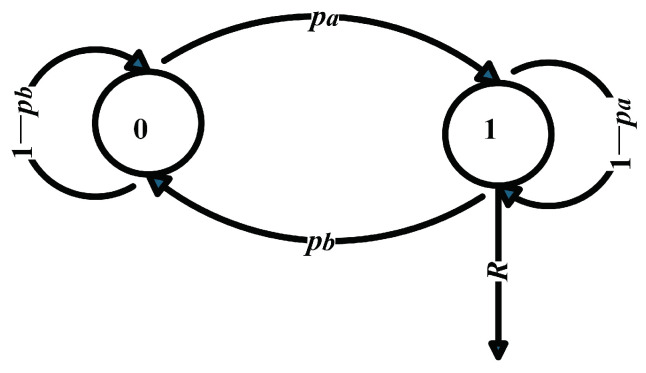
The MMOO model.

**Figure 4 sensors-25-01164-f004:**
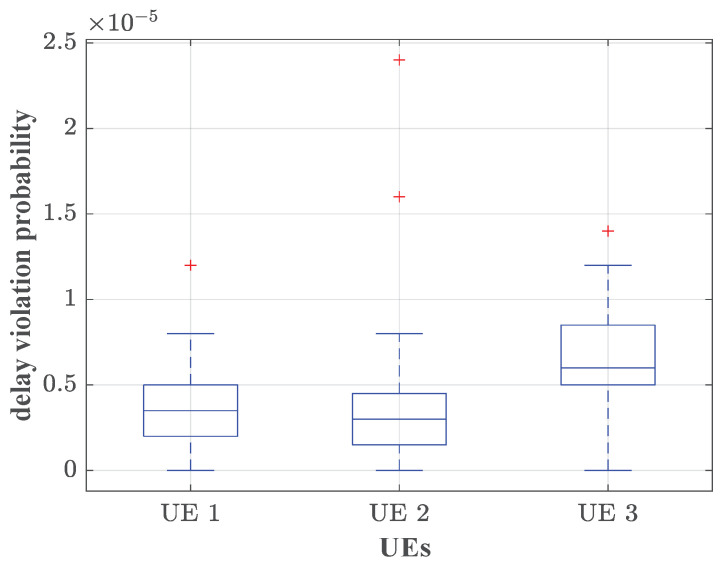
The simulated delay violation probabilities of different pieces of UE, where λ=3 packets/slot, $\varepsilon =10^{-5}$ and R=10 packets/slot.

**Figure 5 sensors-25-01164-f005:**
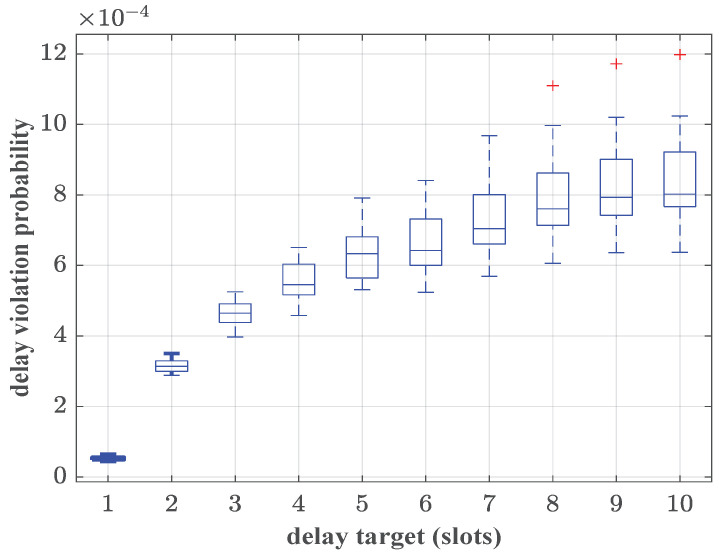
The simulated delay violation probability results, where λ=3 packets/slot, R=4 packets/slot, and $\varepsilon =10^{-3}$. The number of observed time slots is $10^{6}$. For each delay target, 100 simulations are conducted.

**Figure 6 sensors-25-01164-f006:**
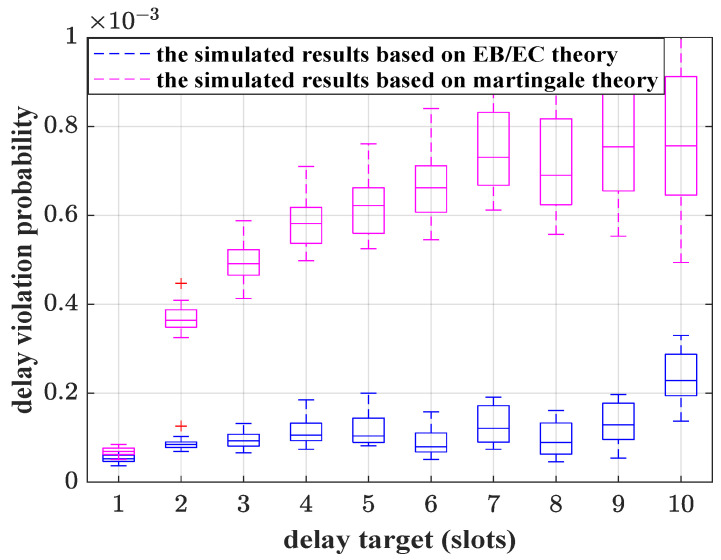
The simulated delay violation probability results based on martingale and EB/EC theories, where λ=3 packets/slot, $\varepsilon =10^{-3}$, and R=4 packets/slot. The results based on martingale theory are purple, and those based on EB/EC are blue.

**Figure 7 sensors-25-01164-f007:**
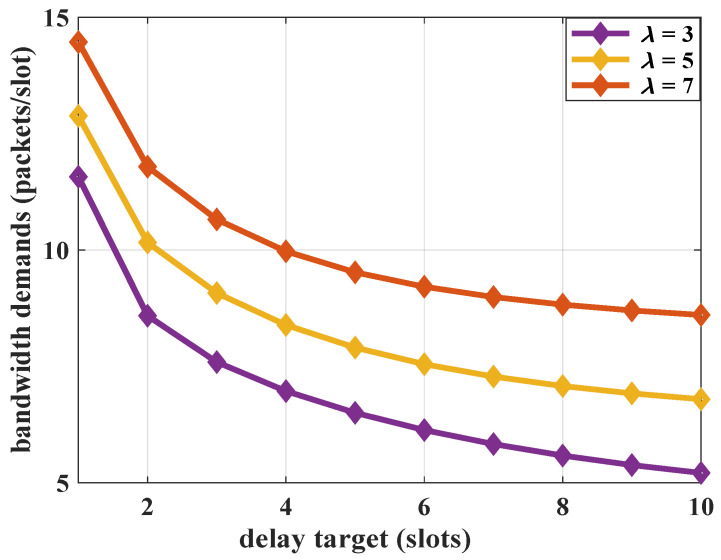
The impact of arrival load on the bandwidth demands, where $\varepsilon =10^{-3}$ and R=5 packets/slot.

**Figure 8 sensors-25-01164-f008:**
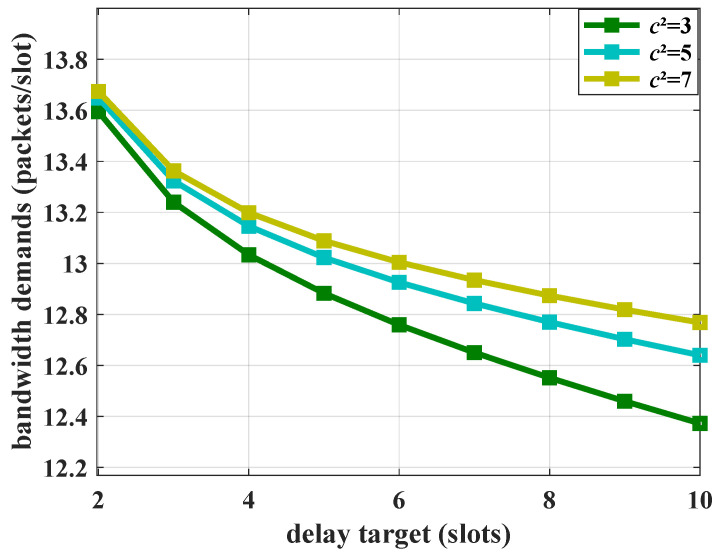
The impact of burstiness on the bandwidth demands, where R=10 packets/slot, λ=3 packets/slot, and $\varepsilon =10^{-3}$.

**Figure 9 sensors-25-01164-f009:**
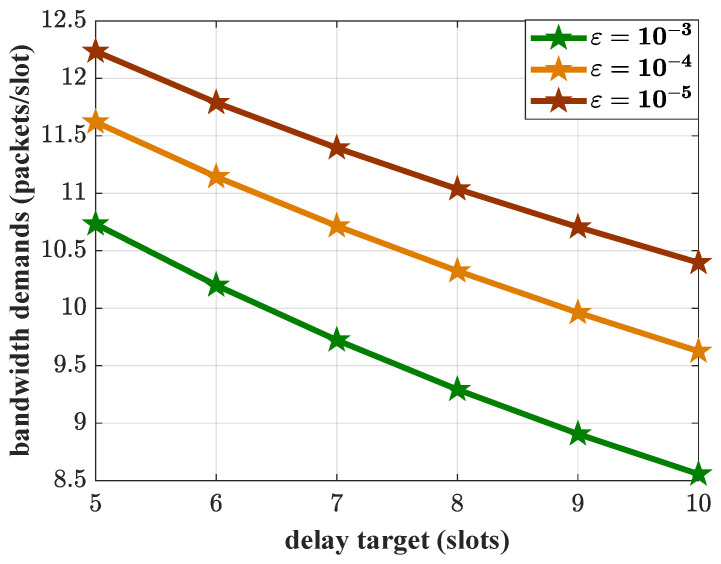
The bandwidth demands vs. delay targets with different delay violation probability thresholds, where λ=3 packets/slot and R=10 packets/slot.

**Figure 10 sensors-25-01164-f010:**
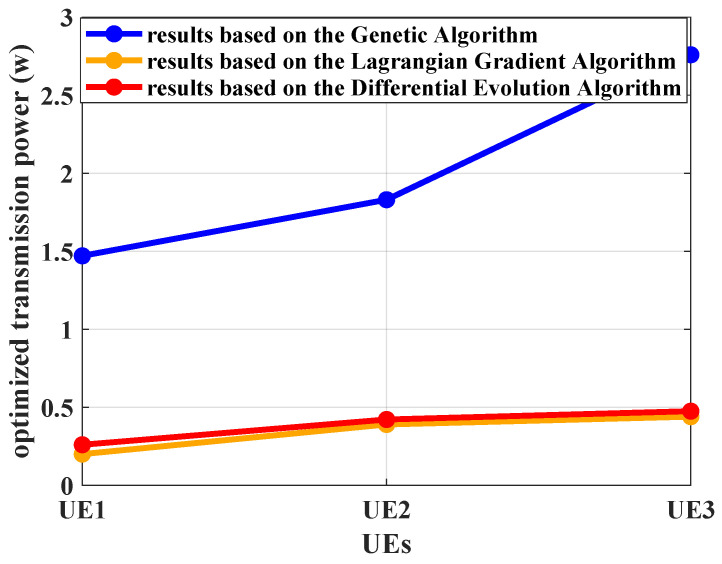
The transmission power of each piece of UE obtained by the Genetic Algorithm, the proposed Algorithm 2, and the Differential Evolution Algorithm, where λ=3 packets/slot, R=10 packets/slot, $\varepsilon =10^{-5}$, and K=3. The distances from each piece of UE to the RRH is Dr=[500,550,600] meters.

**Figure 11 sensors-25-01164-f011:**
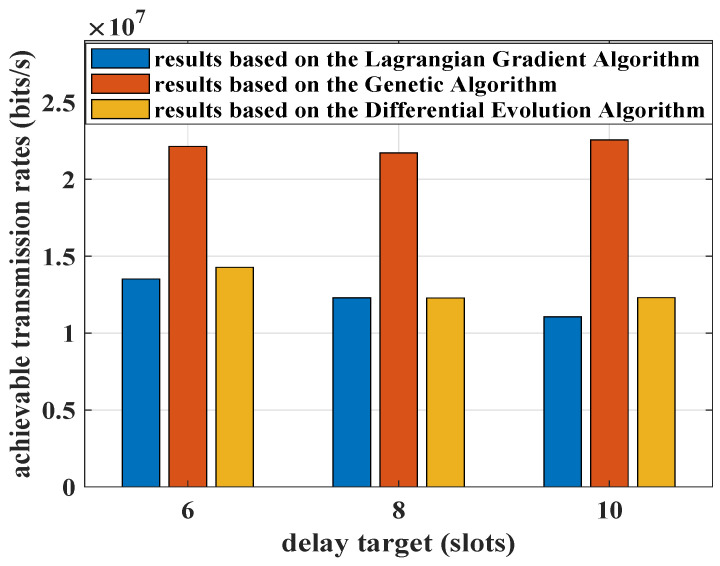
The total achievable transmission rates based on the Genetic Algorithm, the proposed Algorithm 2 and the Differential Evolution Algorithm, where λ=3 packets/slot, R=10 packets/slot, $\varepsilon =10^{-5}$, and K=3. The distances from each piece of UE to the RRH is Dr=[500,550,600] meters.

**Figure 12 sensors-25-01164-f012:**
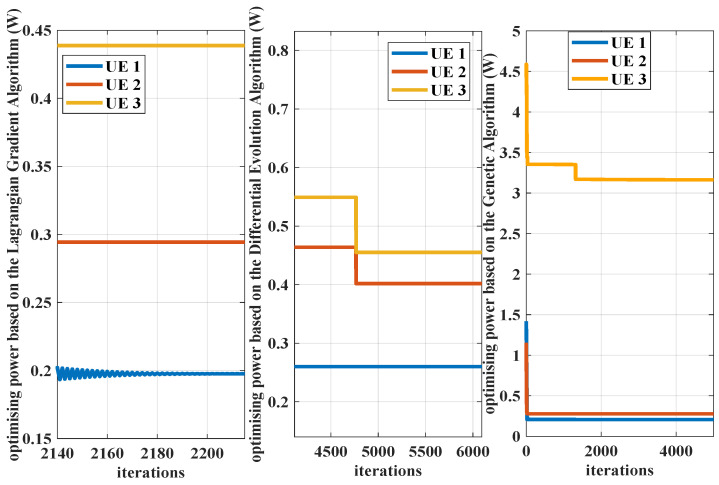
The iterative processes of the Genetic Algorithm, the proposed Algorithm 2 and the Differential Evolution Algorithm, where λ=3 packets/slot, R=10 packets/slot, $\varepsilon =10^{-5}$, and K=3. The distances from the UEs to the RRH are Dr=[500,550,600] meters.

**Figure 13 sensors-25-01164-f013:**
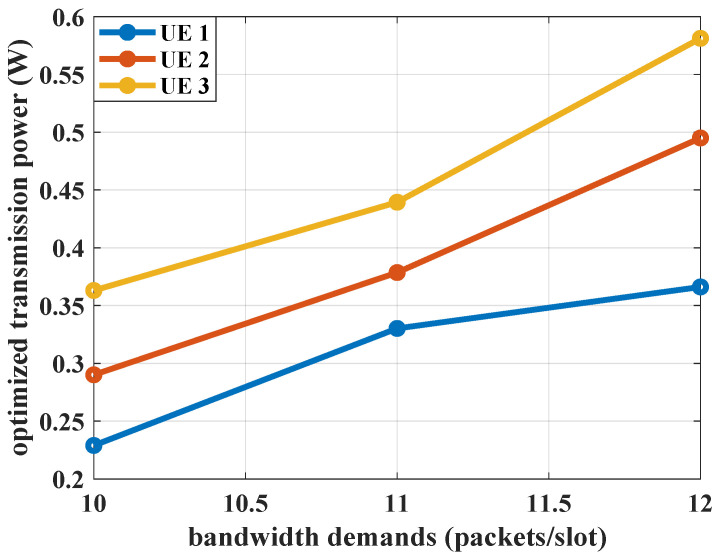
The impact of different bandwidth demands on transmission power, where K=3. The distances from the pieces of UE to the RRH are Dr=[500,550,600] meters.

**Figure 14 sensors-25-01164-f014:**
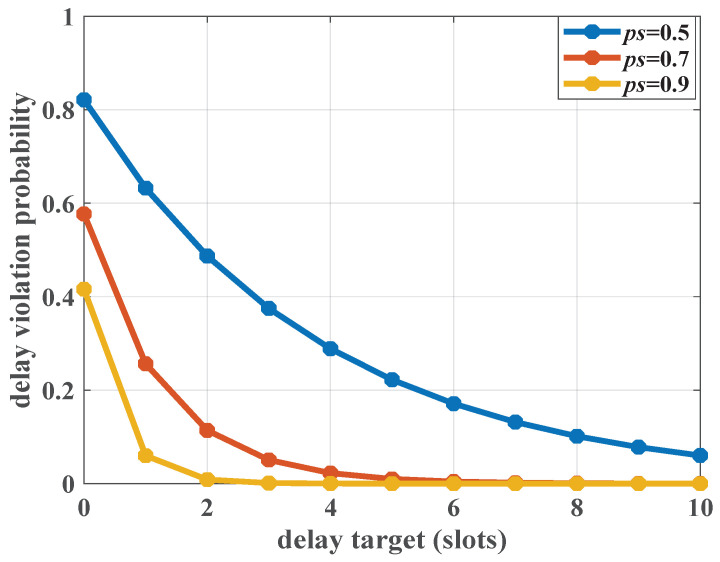
The impact of the successful access probability on delay performance, where λ=3 packets/slot, R=10 packets/slot, and $\varepsilon =10^{-3}$, C=20 packets/slot.

**Table 1 sensors-25-01164-t001:** Comparison between different theory methods in terms of delay performance analysis.

	The analysis of average delay	The analysis of delay violation probability for simple flows	The analysis of delay violation probability for bursty flows	The analysis of delay violation probability for aggregated traffic
Queuing theory	Yes	No	No	No
SNC theory	No	Yes	Yes (but the upper bound is loose and the models of flows are confined)	No
EB/EC theory	No	Yes	Yes (but the upper bound is loose and the models of flows are confined)	Yes (but the upper bound is loose and the models of flows are confined)
Martingale theory	Yes	Yes	Yes (the models of flows are confined)	Yes (the models of flows are confined)

**Table 2 sensors-25-01164-t002:** The commonly used parameters.

Parameters	Values
the duration of a slot *T*	0.125 ms
the bandwidth allocated to a UE *w*	1440 kHz
the maximum transmission power $p_{\max}$	10 W
the delay violation probability threshold ε	$10^{-5}$
the length of a packet *H*	512 bits
the state transition probability $p_{a}$	0.1
the state transition probability $p_{b}$	0.5

**Table 3 sensors-25-01164-t003:** The gap between the simulated delay violation probability and threshold.

$W_{\max}$ (Slots)	1	2	3	4	5	6	7	8	9	10
The gap between the martingale- based result and ε ($\times 10^{-3}$)	0.941	0.639	0.512	0.407	0.344	0.287	0.287	0.264	0.208	0.219
The gap between the EB/EC-based result and ε ($\times 10^{-3}$)	0.961	0.913	0.900	0.888	0.883	0.901	0.828	0.921	0.885	0.785

## Data Availability

The data presented in this study are available upon request from the corresponding author.

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
