# Peer review of "A Delay Performance Analysis and Wireless Resource Allocation Scheme Based on Martingale Theory"

_sensors, 2025, doi:10.3390/s25041164_

Round 1

Reviewer 1 Report

Comments and Suggestions for Authors

lacks the expression or meaning of ALOHA

 The manuscript does not provide sufficient explanation or discussion about the ALOHA protocol, which could be relevant given its foundational role in managing multiple access scenarios in communication systems. Considering the context of aggregated traffic with bursty and i.i.d. flows, the authors should clarify whether and how ALOHA or similar multiple access protocols are incorporated or could influence the proposed delay performance analysis framework. Explicitly addressing this aspect would provide a more comprehensive view of the system's behavior under realistic traffic conditions.

----
Authors should explicitly address the limitations of the study, particularly in terms of its applicability to real-world 5G and emerging 6G network scenarios. For instance, the assumptions of Markov-modulated and i.i.d. traffic might not fully capture the complexity of heterogeneous traffic patterns in practical deployments. 

Reviewer 2 Report

Comments and Suggestions for Authors

This paper proposes a precise delay performance analysis framework based on martingale theory and designs a wireless resource configuration scheme. For aggregated traffic composed of bursty flows and independent identically distributed (i.i.d.) flows, a martingale of aggregated arrival processes is constructed. Based on the definition of martingales, the martingale parameters of Markov-modulated arrivals and i.i.d. arrivals are determined, which exposes the impacts of the heterogeneity of flows on system delay. The paper is interesting, but revisions are definitely required.

1. It is not clear why the authors employed the Lagrangian convex optimization framework. Please explore the preference for Lagrangian convex optimization over other existing frameworks.

2. The authors need to explain how the current work can facilitate achieving a reliability of 99.999% with a delay of milliseconds.

3. It is unclear how equation 1 evolved from Figure 1 based on the Shannon theory to determine the achievable transmission rate of the UE k.

4. In Figures 3, 4, and 7, the authors used the hyphen instead of the minus sign. The two are different. The authors must use the right minus sign. 

5. The performance evaluation of the proposed model is limited. The authors need to examine the computational complexity and overhead costs of the proposed scheme and compare the results with the baseline models.

6. A tabular summary of the key results compared with results from related works is required.

7. The following reference materials would be suitable for improving the quality of the paper. 1. Behera, J.R.; Imoize, A.L.; Singh, S.S.; Tripathy, S.S.; Bebortta, S. Optimizing Priority Queuing Systems with Server Reservation and Temporal Blocking for Cognitive Radio Networks. Telecom 2024, 5, 416-432. https://doi.org/10.3390/telecom5020021. 2. Du, Y., Yang, L., & Luo, Y. (2025). Resource allocation based on optimized cellular network AP layout for visible light communication heterogeneous network. The Journal of Supercomputing, 81(1), 1-23. 

8. Minor English editing of the paper is required.

Reviewer 3 Report

Comments and Suggestions for Authors

1, The introduction lacks depth in discussing the specific technical challenges and current state-of-the-art in URLLC applications. It is essential to provide a more comprehensive and up-to-date literature review to establish a solid foundation for the research. This section should clearly highlight the gaps in the existing literature that this study aims to address.
2,
The research objectives are somewhat vague, and the contributions of this work are not distinctly highlighted. The authors must clearly define the primary and secondary objectives and explicitly state the unique contributions of this study compared to existing research.
3,
The methods section, while detailed, lacks clarity in some areas. The authors must provide a more thorough description of the assumptions made and the limitations of the models used. This will enhance the transparency and reproducibility of the research. Additionally, the authors should include a step-by-step explanation of the algorithms and mathematical models used, ensuring that they are accessible to a broader audience.
4,
The discussion section is underdeveloped. The authors must expand this section to include a more in-depth analysis of the results, comparing them with those of previous studies and discussing any discrepancies. The authors should also discuss the practical implications of the findings and any potential limitations of the study.
5,
While the English is generally understandable, there are several grammatical and stylistic issues that detract from the clarity and professionalism of the manuscript.There are inconsistencies in the labeling of mathematical equations, figures, and tables. The authors must double-check all these elements to ensure consistency and accuracy.
6,
The paper lacks practical examples or case studies that illustrate the application of the proposed methods in real-world scenarios. The authors must include at least one practical example or case study to demonstrate the practical relevance and applicability of the research.

Reviewer 4 Report

Comments and Suggestions for Authors

This paper proposed a precise delay performance analysis framework based on martingale theory and a wireless resource configuration scheme. But there are some problems in this paper. There are three contributions in this paper. The first is that a martingale for aggregated arrival processes is proposed. The second is that a precise upper bound of delay violation probability is derived. And the third is that a bandwidth estimation algorithm is proposed. And there two parts in this paper, the first part is the theory analysis by using martingale, and the second part is the resource allocation scheme. But the relationship between these two parts is not clear, please explain it.

Reviewer 5 Report

Comments and Suggestions for Authors

The paper constructs a martingale process of aggregated traffic based on martingale theory and proposes a novel delay performance analysis framework and wireless resource allocation scheme, which shows certain innovation. When handling aggregated traffic comprising bursty flows and independent identically distributed flows, it discloses the influence of heterogeneous flows on system delay by determining martingale parameters, presenting a unique perspective in related research. Moreover, the upper bound of the delay violation probability obtained using the martingale stopping time theory is tighter than existing methods, providing more effective theoretical support for resource allocation in Ultra-Reliable Low Latency Communications (URLLC). However, there are still some issues that need to be properly addressed.

  1. The introduction does not fully cite the current state of the field. When introducing queuing theory, effective bandwidth and capacity theories, and martingale theory, can specific application cases or real-world data be added to provide a clearer comparative explanation of martingale theory in solving delay QoS issues? Consider listing specific challenges encountered when using other methods in real network scenarios and how martingale theory can effectively address these issues.

  2. In the discussion of the research background, can scenarios related to the development of future communication networks be added, especially considering the maturity and wide application of 5G technology? This would highlight the paper's cutting-edge and innovative aspects. For example, applications in industrial IoT and intelligent transportation could be emphasized, along with the significance of this research in advancing these fields.

  3. There are not enough figures and charts. When describing the network model and queuing system, could some diagrams or schematic illustrations be added? These would visually represent the distribution relationships between RRH and UE, as well as the data transmission and queuing process. For instance, it would be ideal to include a parameter table alongside comparative graphs.

  4. Some characters in the equations are not explained, and the formatting is not aesthetically pleasing. For example, in the explanation of Equation (7), when the denominator is relatively simple, a slash (/) could replace the fraction. Line breaks in equations should also be avoided whenever possible.

  5. Regarding the description of path loss, shadow fading, and small-scale fading in the channel gain model, further explanations of how these factors specifically impact transmission rate and delay performance would help readers better understand the physical significance of the model.

  6. When defining the martingale process for aggregate arrival and service processes, the parameters and functions in the equations could be explained in more detail and with greater clarity. Simple examples or analyses of special cases could help readers better understand how these parameters are determined and their roles in the martingale process.

  7. In describing the experimental setup, details about the simulation environment, such as the software or platform used, could be included. Additionally, information on the simulation's time scale and accuracy would enable other researchers to better reproduce the experiments.

  8. For the analysis of experimental results, statistical tests such as significance tests could be conducted to rigorously verify the effectiveness of the proposed approach. Moreover, comparisons with other existing methods could be added to more intuitively demonstrate the advantages of this research.

  9. In the overall structure of the paper, the transitions and coherence between sections could be further optimized. For instance, in the process from theoretical analysis to algorithm design and experimental validation, transitional paragraphs could be added to explain the logical relationships and research progression between each section.

  10. The references cited in the paper appear only in the introduction, which does not align with general academic writing practices. Consider citing references for specific methods or terms used throughout the paper to make it more persuasive.

  11. In terms of language expression, the professionalism and accuracy of the paper can be further improved. Ambiguous expressions should be avoided. Additionally, a professional English editor or native English speaker could be invited to polish the paper's language to improve its readability.

Comments on the Quality of English Language

The overall vocabulary is relatively professional and meets the requirements of an academic paper. The use of specialized terminology in the field of communication studies, such as "martingale," "Ultra-Reliable Low Latency Communications (URLLC)," and "queuing theory," is relatively accurate and effectively conveys the core concepts of the research. However, the use of some non-specialized words can be further refined. In some sentences, there are relatively common or conversational expressions. Replacing these with more formal and precise terms could enhance the paper's professionalism.

Round 2

Reviewer 2 Report

Comments and Suggestions for Authors

Dear Authors,

Thank you for addressing my comments.

Author Response

    Thank you very much for your precious time and efforts expended in helping improve our paper.  

    In this version, we have improved the Abstract, Introduction and Contributions to highlight the work. The proposed theoretical analysis method and the designed algorithms have been clarified more clearly. More results and analysis have been added in the manuscript to reveal the performance of the proposed delay QoS guarantee scheme. All the modifications and discussions have been marked red in the revised manuscript. We attempted to address all of your concerns. Thanks again for your positive comments on our work.

    Sincerely,

    The Authors

Reviewer 5 Report

Comments and Suggestions for Authors

The paper "A Delay Performance Analysis and Wireless Resource Allocation Scheme Based on Martingales" focuses on the research of delay performance analysis and wireless resource allocation in Ultra - Reliable Low - Latency Communications (URLLC) scenarios. The selected topic aligns with the development needs of the current communication field, and it holds high research value and practical significance. The article has a complete structure and clear logic. From theoretical analysis, algorithm design to experimental verification, each part is closely connected, demonstrating the author's solid professional knowledge and rigorous scientific research attitude in this research direction. By constructing a unique analysis framework based on martingale theory and proposing corresponding bandwidth estimation and resource allocation algorithms, the research results show certain innovation and play a positive role in promoting the development of URLLC technology. After careful review, it is recommended to accept the paper after making revisions on the following minor issues.

-----
The bandwidth estimation algorithm uses a binary search method combined with the effective bandwidth theory to determine bandwidth requirements. This approach can meet the requirements to some extent, but the accuracy control during the search process can be further optimized. For example, when setting the termination condition for the binary search, the current method only considers the difference between the theoretical delay upper bound and the target threshold. This may result in the final determined bandwidth not being the optimal solution. It is recommended that the author consider introducing an additional precision control parameter. For instance, the search could be terminated when the bandwidth change between consecutive iterations is smaller than a very small predefined value, thereby improving the accuracy of bandwidth estimation and more precisely meeting delay QoS requirements.

In the experimental section, the authors compare the transmission power of the proposed algorithm with that of the genetic algorithm, but the comparison is relatively limited in scope. To comprehensively evaluate the algorithm's performance, it is recommended to include additional performance metrics, such as system throughput and spectral efficiency. These metrics are crucial for assessing the overall performance of wireless resource allocation algorithms in real-world networks. A multi-dimensional comparison would provide a clearer presentation of the proposed algorithm's strengths and weaknesses, offering readers more comprehensive insights.

Comments on the Quality of English Language

The paper contains a large number of technical terms. While some terms are clearly explained upon their first appearance, inconsistencies exist in their usage and explanations in later sections. For example, in different formula derivations and theoretical analysis sections, the definition and citation of the "martingale process" vary slightly, which may cause confusion for readers. It is recommended that the authors standardize the explanations and usage of technical terms, providing a complete and accurate definition upon first mention and maintaining consistency throughout the paper. This will enhance readability and improve the logical coherence of the manuscript.

Figures and tables play a crucial role in presenting experimental results and data. However, some figures lack complete labeling information. For instance, the meanings of coordinate axis labels are not always clearly indicated, and some legends are too brief. Improving the labeling details would allow readers to interpret the data more intuitively and accurately, thereby enhancing the efficiency of information delivery in the paper. The authors should carefully review and refine the labeling of all figures to ensure completeness and accuracy
